# Balancing Mission and Margins: What Makes Healthy Community Food Stores Successful

**DOI:** 10.3390/ijerph19148470

**Published:** 2022-07-11

**Authors:** Sara John, Megan R. Winkler, Ravneet Kaur, Julia DeAngelo, Alex B. Hill, Samantha M. Sundermeir, Uriyoan Colon-Ramos, Lucia A. Leone, Rachael D. Dombrowski, Emma C. Lewis, Joel Gittelsohn

**Affiliations:** 1Center for Science in the Public Interest, Washington, DC 20005, USA; 2Department of Behavioral, Social, and Health Education Sciences, Rollins School of Public Health, Emory University, Atlanta, GA 30322, USA; megan.winkler@emory.edu; 3Division of Health Research and Evaluation, Department of Family and Community Medicine, University of Illinois College of Medicine, Rockford, IL 61107, USA; ravneetk@uic.edu; 4Departments of Health Policy Management & Nutrition, Harvard T.H. Chan School of Public Health, Harvard University, Boston, MA 02115, USA; jdeangelo@hsph.harvard.edu; 5Urban Studies and Planning and Detroit Food Map Initiative, Wayne State University, Detroit, MI 48202, USA; alexbhill@wayne.edu; 6Department of International Health, Johns Hopkins Bloomberg School of Public Health, Johns Hopkins University, Baltimore, MD 21205, USA; srex2@jh.edu (S.M.S.); elewis40@jhu.edu (E.C.L.); jgittel1@jhu.edu (J.G.); 7Milken Institute School of Public Health, George Washington University, 950 New Hampshire Avenue, Washington, DC 20052, USA; uriyoan@gwu.edu; 8Department of Community Health and Health Behavior, University at Buffalo, Buffalo, NY 14260, USA; lucialeo@buffalo.edu; 9Division of Kinesiology, Health and Sport Studies, College of Education, Wayne State University, Detroit, MI 48202, USA; fy9585@wayne.edu

**Keywords:** food access, nutrition, healthy food retail, retail food environment, community store, store success, urban, case study approach, cross-case analysis

## Abstract

Mission-driven, independently-owned community food stores have been identified as a potential solution to improve access to healthy foods, yet to date there is limited information on what factors contribute to these stores’ success and failure. Using a multiple case study approach, this study examined what makes a healthy community food store successful and identified strategies for success in seven community stores in urban areas across the United States. We used Stake’s multiple case study analysis approach to identify the following key aims that contributed to community store success across all cases: (1) making healthy food available, (2) offering healthy foods at affordable prices, and (3) reaching community members with limited economic resources. However, stores differed in terms of their intention, action, and achievement of these aims. Key strategies identified that enabled success included: (1) having a store champion, (2) using nontraditional business strategies, (3) obtaining innovative external funding, (4) using a dynamic sourcing model, (5) implementing healthy food marketing, and (6) engaging the community. Stores did not need to implement all strategies to be successful, however certain strategies, such as having a store champion, emerged as critical for all stores. Retailers, researchers, philanthropy, and policymakers can utilize this definition of success and the identified strategies to improve healthy food access in their communities.

## 1. Introduction

Limited healthy food access is a contributing driver to nutrition insecurity and nutrition-related health disparities (see Appendix A for definitions of terminology used throughout the paper). Forty million Americans live in areas defined by the United States Department of Agriculture (USDA) as low-income and low-food-access, often referred to as “food deserts” [1,2]. In recent years, the term “food apartheid” has been used to acknowledge structural barriers to food, which are rooted in racism and classism, intentional, and not naturally occurring [3]. Indeed, chain supermarkets are less likely to locate in neighborhoods with lower incomes [4].

Placing supermarkets in historically marginalized communities with limited food access has been a popular solution in the U.S. In 2010, the Obama Administration announced $400 million for the Healthy Food Financing Initiative (HFFI) to incentivize food retailers to locate in underserved urban areas and rural communities, and HFFI received additional funding in the past two farm bills [5,6,7]. However, subsequent evaluations of supermarket entry into low-income, low-access areas found little to no improvement in associated healthy food availability, purchases, or consumption among neighborhood residents [8,9,10,11].

An alternative strategy has been to leverage the power within communities to improve the community food environment through independently-owned community food stores. Challenges exist with this approach. Nontraditional retailers, including small grocery stores, corner stores, and markets, that serve an outsized role in communities with limited economic resources tend to stock few healthy items and sell foods at higher prices than supermarkets [12,13,14]. These trends are unsurprising given the numerous obstacles smaller stores face to stock affordable healthy food, including inadequate distribution channels, limited purchasing power, supplier-mandated merchandising, and broader capacity challenges [15,16,17,18].

Despite these obstacles, some community food stores have improved healthy food access through healthy retail pilots. Previous intervention evaluations indicate community food stores can use healthy food product, placement, promotion, and pricing strategies to improve the store environment [19,20,21,22,23], increase healthy food purchases [20,23,24,25,26], and even improve customer health outcomes [27,28]. However, we lack evidence on if and how healthy retail marketing interventions may be integrated into community food store business models, strategies beyond marketing that can be adopted to improve health food access, and subsequent implications for store success.

Some community food stores across the country have adopted missions to increase healthy food access and thus provide an opportunity to gain insights from these distinct retailers. These mission-driven stores work through for-profit, social enterprise, cooperative, and non-profit business models to meet food needs and to create economic opportunities for the communities in which they reside, often incorporating public health, environmental, and/or social justice goals [29]. However, no research to date has reported systematic assessments of mission-driven community food stores. Additionally, limited evidence exists on store successes and failures using a contextualized approach that can provide a deeper understanding of why a store is or is not successful. Data limitations in small and independent food retailers contribute to this being an understudied food environment [15], however an approach that triangulates across available qualitative and quantitative data sources would be well-suited for this setting.

In this study, we used a case study approach to explore what makes healthy community stores successful or unsuccessful at making healthy food affordable to customers with limited economic resources while maintaining a viable business, as well as what strategies can help or hinder these goals. Case studies excel at placing a specific case (i.e., a community food store) within its context, and case study analysis is one of the strongest analytic approaches for maintaining context [30]. By gaining detailed insights into these stores, we may identify how other food retailers could increasingly move toward offering affordable, high-quality, healthy foods in their communities.

## 2. Materials and Methods

### 2.1. Study Design

This paper is one in a series of papers from the Healthy Community Stores Case Study Project, which is funded by Healthy Eating Research, a national program of the Robert Wood Johnson Foundation, and in collaboration with the Centers for Disease Control and Prevention Nutrition and Obesity Policy Research and Evaluation Network (NOPREN) Food Retail Work Group [31]. The larger project aimed to understand how independently-owned community food stores increase healthy food access in economically vulnerable communities, as well as how community food stores were impacted by the COVID-19 pandemic and racial uprisings of 2020. The project used a multiple case study design and analysis to obtain a highly-contextualized understanding of seven healthy community food stores [32], with study protocol further detailed in Gittelsohn et al., 2022 also in this special issue [31]. This paper specifically utilizes multiple case study analysis to understand what makes a healthy community food store successful and the strategies and contexts associated with store success. See Appendix B for a study methodological flow diagram.

### 2.2. Recruitment and Data Collection

#### 2.2.1. Recruitment

As further detailed in the study protocol paper, the seven community food stores featured in this study were recruited through a self-nomination process conducted through the NOPREN Food Retail Work Group [31]. To be considered for inclusion in the study, stores had to: (1) serve low- or low-to-middle income communities, (2) be open for at least one year, (3) have a mission statement to improve healthy food access, (4) be willing to share procurement, stocking, and sales data, and (5) be a Supplemental Nutrition Assistance Program (SNAP) and/or the Special Supplemental Nutrition Program for Women, Infants, and Children (WIC) authorized retailer. Purposive sampling was used to strategically choose information-rich cases. Seven community food stores were selected using a maximum variation sampling approach that maximized diversity on dimensions of interest [33], including store type (e.g., grocery store, corner store), business model (e.g., for-profit, non-profit), and geographic location (e.g., Northeast, Midwest) [31].

#### 2.2.2. Data Collection

Multiple data sources were collected at each site to inform the individual case reports, including in-depth interviews (IDIs), the Nutrition Environment Measures Survey for Stores (NEMS-S) short form, store websites, articles published about the stores, and supplementary materials shared by store staff.

IDIs with store stakeholders were a critical data source for the project. The research team conducted IDIs with 3–4 staff per store, including store owners, managers, and employees, as well as 3–4 external stakeholders per store, including community members, non-profits, foundations, and vendors. The use of multiple perspectives, including those primary and secondary to the store, in combination with multiple data sources strengthened within-case triangulation [32].

Quantitative store environment assessments were another key data source. The NEMS-S is a reliable tool with the capacity to reveal significant differences in healthy food availability and pricing across store and neighborhood food environments [34]. A modified, shorter version of the NEMS-S was utilized [35,36,37,38]. The NEMS-S short form assessed healthy food availability, quality, and price, and it was conducted three times per store.

Additional data collection details are available in the study protocol paper [31].

### 2.3. Analysis

Analysis for this study used the individual case reports (described below) as well as more detailed quantitative data collected via the NEMS-S that was not included in the original case reports. The analytic approach to the NEMS-S quantitative data collected at each site is presented first, followed by the case analysis approach used to generate and to analyze the case reports for each site.

#### 2.3.1. Healthy Food Availability Healthy Eating Index (HFAHEI)

The USDA’s Healthy Eating Index (HEI) is the standard for measuring healthful eating by Americans, yet there is no standard measure for food availability. The Healthy Food Availability for Healthy Eating Index (HFAHEI) was developed as a part of this project to align the 13 HEI categories with the 11 food categories assessed in the NEMS-S, further detailed in Hill et al., 2022 also in this special issue [39]. This adapted food availability index for healthy eating builds on previous work by Franco et al. that utilized the NEMS-S availability sub-score and similar modified NEMS measures for healthy food availability by Campbell et al. [40,41]. The HFAHEI aligns 10 categories across the HEI and NEMS-S to get a clearer sense of the local opportunity for healthy eating based on healthy food availability. The HFAHEI scores range from 0 to 30 with higher scores indicating greater availability of healthful food items that align with the HEI.

#### 2.3.2. Food Affordability Indicator

Pricing of healthy food is a commonly cited barrier to healthful eating [42,43]. Pricing data collected from all participating food retailer locations were analyzed against the Consumer Price Index (CPI) for select items to compare how case study sites followed national trends in food pricing [44]. Food items from the CPI were limited to healthy food items assessed as part of the NEMS-S. Five food categories from the NEMS-S matched six available food items in the CPI for comparison. The food items for comparison included: low-fat milk, apples, bananas, tomatoes, chicken breast, and wheat bread. The NEMS-S was predominantly implemented over May–July 2021. The case study sites’ food prices were averaged across those months, and similarly a three-month average of the CPI was pulled for comparison.

#### 2.3.3. Individual Case Reports

Each site began by analyzing their multiple data sources, including coding qualitative data and calculating the HFAHEI as described above, to create a within-case report that followed an outline put forth by the multi-site study team. These reports provided highly descriptive accounts documenting the unique experiences each store had as they aimed to make healthy foods affordable in their communities and navigated the events of 2020 (e.g., COVID-19 pandemic). Key sections of the case reports included store setting, mission, organizational structure, sourcing, stocking, pricing, community engagement, and COVID-19 adaptations. During individual case report writing, each site had multiple insider perspectives evaluate the conclusions drawn from the multiple data sources [32].

#### 2.3.4. Stake Multiple Case Study Analysis

Stake’s multiple case study analysis approach, which moves from within- to cross-case analysis, was used in this study [32]. After the completion of individual case reports, the analysis was led by a lead analyst (SJ) and moved to the cross-case phase, following Stake’s multiple case study analysis approach [32]. This analysis maintained the contextual situatedness of each case, while investigating for similarities and differences across cases. First, the lead analyst read each within-case report to develop great familiarity with each case, and they identified findings relevant to stores making healthy food accessible. The analyst also assessed each case for its overall usefulness in answering the research question, which allowed both cases with prominent data on making healthy food affordable as well as atypical cases with uncommon experiences to be identified.

Next, the lead analyst conducted sorting and merging activities outlined by Stake to identify cross-case findings, while still maintaining case context, including sorting findings based on similarity and ranking merged findings to assess relative importance in answering the research question [32]. These steps served as the foundation for developing meaningful interpretations around the success of making healthy food affordable in the multi-case study as a whole as well as the contextual factors that led to a case’s success.

During the cross-case phase, the lead analyst frequently referenced and re-read case reports as well as performed data checks and confirmations with each site about any questions in the respective case report. Analysis was also supported by a smaller group of co-authors (MW, RK, JD) familiar with the cases that provided frequent feedback on the extracted case findings, merged findings, and interpretations. Cross-case triangulation included routine and spontaneous discussions with the broader working group, analysis subgroup, and site-specific project leaders to increase the accuracy, comprehensiveness, and clarity of findings and their interpretations [32].

### 2.4. Ethical Considerations

IRB approval was obtained by each site’s associated research institution. The stores and informants interviewed in this project have been deidentified to preserve their anonymity. Throughout the paper, the stores are referred to as the city they are located in (e.g., Baltimore, Boston), however the results are solely representative of the singular store.

## 3. Results

An overview of the seven stores included in the cross-case analysis is presented in Table 1 with store locations mapped in Figure 1. The stores included grocery stores, corner stores, a market, and a supermarket, and they varied in financial models (for-profit, non-profit, and co-op). The stores also ranged in size (900–65,000 square feet) and years opened (3–50 years), with one store closing between project start and analysis (Baltimore, MD, USA).

In the remainder of the Results section, the research team first defines healthy community store success (Section 3.1), then describes the strategies and the contexts that lead to store success or lack of success (Section 3.2), and ends by considering how stores balance missions and margins (Section 3.3).

### 3.1. What Does Success Look like for a Healthy Community Store?

We identified a three-prong definition of success for healthy community stores: (1) providing healthy food, (2) offering healthy food at affordable prices, and (3) reaching community members with limited economic resources. We found that stores displayed variation in their commitment to these areas, ranging from intention to achievement (Table 2).

#### 3.1.1. Store Commitment to Providing Healthy Food

Stores varied in their efforts to prioritize and to define “healthy foods”. All stores signaled that the provision of healthy foods was a store priority by including it in their store mission. However, when it came to putting this intention into action, stores had to weigh their mission against their perceived customer demand. As the Chicago store owner summarized,
*“Let’s just say for instance, I brought in a case of a new product that’s low in fat, low in sugar, and I put it on my shelf. And people are hesitant to buy that product because of the perception that it’s probably less tasty to them… A lot of store owners, a lot of small business store owners are hesitant to even bring in that product from the very beginning for that very reason”.*(Chicago store owner)

Alternatively, other stores considered that their role as part of their mission was to create, not just respond to, customer demand. The Buffalo store owner recommended other stores be persistent with their healthy food offerings,
*“Don’t give up if you don’t sell today…when I started, I started just bringing one banana, one case. Now we sell almost 12, 15 in every week”.*

Similarly, the Boston store founder felt a responsibility as a healthy community store to increase healthier purchases but also to decrease less healthy purchases. He rationalized,
*“The last thing [customers] need is for a spot like [our store] to carry soda and cookies and candy stuff, particularly at a discount. That would be not helpful for our mission”.*(Boston store founder)

Considering these tensions, not all stores’ health missions moved to action. Three stores (Boston, DC, and Minneapolis) defined “healthy” with variation in scope and specificity across sites. Only Boston explicitly defined “healthy food” by creating retail and prepared food nutrition guidelines. For instance, beverages with more than 8 g of sugar per 8-ounce serving and artificially sweetened beverages were not stocked per store guidelines. The other stores did not define “healthy foods” but seemed to use fruits and vegetables as a proxy.

DC and Minneapolis stores defined “healthy” more broadly to encompass the health of the communities, incorporating concepts beyond nutrition. For example, DC emphasized the importance of selling local produce to achieve the dual purpose of increasing fruit and vegetable access and supporting the local economy. Even broader, Minneapolis envisioned a healthy community as one that has “equitable economic relationships, positive environmental impacts, and inclusive, socially responsible practices,” and a store leader acknowledged that,
*“…we don’t specifically say healthful as a criteria, but I think the combination of what we do prioritize, I think results in what many people would describe as healthful… those decisions are very intentional. Our fresh foods are the majority of what we prioritize from the merchandising standpoint”.*(Minneapolis store leader)

Stores with both implicit and explicit definitions of “healthy food” operationalized their health mission through a variety of strategies, including increased stocking and merchandising of these foods. Boston was the only store to stock only items that qualified as “healthy” according to their preestablished nutrition guidelines, whereas the other stores offered a mix of products without nutrition criteria, including sugar-sweetened beverages and energy-dense, nutrient-poor snacks. Ultimately, according to this team’s observations using the HFAHEI scores, all stores stocked some healthy foods with HFAHEI scores ranging from 11.6–27.5, but only the stores located in Boston, DC, Detroit, and Minneapolis had HFAHEI scores over 20. A store with a score over 20 had a large variety of healthy foods available, while lower scoring stores either did not stock certain healthy food categories or stocked fewer of those items.

#### 3.1.2. Store Commitment to Making Healthy Foods Affordable

Stores also recognized the importance of affordability to make healthy foods accessible in their communities, and some stores altered their business models to be able to offer lower prices. All stores acknowledged the necessity to provide foods at affordable prices in communities with limited economic resources. Yet, many stores, especially smaller stores with more limited purchasing power, struggled to provide low prices. A Boston store employee summarized,
*“Because sometimes, you know we’re smaller, we’re not getting a big bulk discount like a larger grocer would, so we don’t have the best costs in town and we’re trying to provide the best prices”.*(Boston store employee)

Affordability was further complicated at stores with more expansive missions beyond supplying nutritious food. Minneapolis strived to provide high-quality, natural, and local products that support a fairer food system, inevitably leading to higher prices. A Minneapolis store leader explained,
*“…the hard rub is that there’s a lot of people in poverty in our country and it’s gotten worse… they can’t afford food that is premised on not having people in poverty. So there is a disconnect there that we…can’t reconcile it. We can’t make the food at the price that they can afford”.*(Minneapolis store leader)

Despite these tensions, most stores (Baltimore, Boston, Buffalo, DC, Minneapolis) altered business practices to provide lower prices to customers with limited economic resources, including through their financial model, sourcing strategies, and innovative external funding (strategies discussed in greater detail in Section 3.2). Ultimately, the prices of healthy food at stores ranged from 194% to 97% of the national average estimated by the Consumer Price Index, with Baltimore, Boston, and Detroit selling food at prices below the national average. A Boston store employee reflected on interactions with customers that underscored the importance of affordable pricing in their community,
*“They’ll ask me, ‘are you sure you rang everything up?’—they can’t believe the amount of food they’re getting for that price. It can bring them to tears”.*(Boston store employee)

#### 3.1.3. Store Commitment to Reaching Community Members with Limited Economic Resources

Finally, all stores understood the importance of connecting with the communities they served, and most took action to engage with community members with lower incomes and provide resources beyond those of a traditional grocery store. Establishing credibility through community engagement was critical for those stores with leadership that lived outside the community (Baltimore, Boston, Chicago, DC, Detroit,). A Detroit customer explained why community engagement is critical for stores in communities with limited economic resources,
*“I feel like a lot of the mistrust with [the supermarket], too, is that you know just in the past, a lot of grocery stores in the neighborhood have not had that quality produce, quality products, just in general. And to be able to, you know, make sure people establish a trust issue, I feel like doing stuff with the kids, doing stuff with the youth, doing stuff more community based, and can kind of help, you know, instill that trust”.*(Detroit store customer)

In contrast, store owners who were community members themselves, such as Buffalo and Minneapolis, were inherently connected with the community, a topic discussed in greater detail in Dombrowski et al., 2022 also in this special issue [45].

All stores except for Baltimore successfully engaged with community members, including through work with community-based organizations (CBOs) and existing community programs, although the level of community involvement varied across stores, further detailed in Kaur et al., 2022 also in this special issue [46]. Ultimately, Boston, Buffalo, Chicago, DC, Detroit, and Minneapolis reached economically vulnerable community members, with their customer base reflecting the makeup of the broader community. In the words of DC store management,
*“Our customer mix is fairly indicative of the demographics of our community and that is our goal. I mean we really want our store to look and feel like our community”.*(DC store manager)

### 3.2. What Strategies Lead to Healthy Community Store Success?

We also identified strategies for *how* these stores can successfully make healthy food accessible to communities with limited economic resources. We found six strategies for success: (1) having a store champion, (2) using nontraditional business strategies, (3) obtaining innovative external funding, (4) using a dynamic sourcing model, (5) implementing healthy food marketing, and (6) engaging the local community. Across all strategies, we identified a key pattern that healthy community stores’ owners and staff perceive: they must weigh their responsibilities both to their business and to their community. Regardless of their financial model, stores needed to stock, price, and sell healthy items to benefit their bottom line but also meet their communities’ needs. Several distinct strategies and contexts were identified as facilitating a store’s ability to successfully balance mission and margins (Table 3). While stores did not need to adopt all strategies to be successful, a certain threshold needed to be met, and specific strategies emerged as critical.

#### 3.2.1. Having a Store Champion

Store leadership served as a critical key champion for some stores’ success, providing expertise and multi-faceted roles spanning business and community, also discussed in Dombrowski et al., 2022 in this special issue [45]. Existing research has explored the importance of community business owners in health promotion through barbershops and beauty salons [47,48,49,50] and found business owners’ social entrepreneurship and community health values as key factors for success [47,48,49].

Additionally, the role of champions differed for stores with origins within the community (e.g., Buffalo and Minneapolis) from stores with origins external to community (e.g., Boston and DC). Community-led stores had innate community representation, whereas stores with external leadership needed to take additional steps to authentically and meaningfully engage community members, a topic explored more deeply in Kaur et al., 2022 [46]. In Minneapolis, community members owned and democratically governed the store through their food cooperative or co-op model; co-op owners advocated for improved healthy food access by navigating what is good for the store business, local food system, and community. In Buffalo, the store owner was a longtime community resident and motivated to juggle many responsibilities to increase healthy food access. A Buffalo community stakeholder said of the store owner,
*“He’s constantly trying to expand his reach; he’s constantly trying to support more folks, whether it is through his store or just within the community”.*(Buffalo store owner)

Other charismatic store champions lived outside the community but utilized knowledge across retail operations, philanthropic funding, and food access issues to support stores. The Boston founder had decades of experience in the grocery business and a deep understanding of the potential role a mission-driven retailer could play to support healthy food access. He leveraged his many industry, philanthropy, and academic connections to develop and to refine a non-profit grocery store that provided healthy foods at low prices. A funder of the Boston store stated,
*“If [store founder] couldn’t do it, no one could do it. He was basically trying to leverage everything that he had learned and everyone that he knew in order to make [the store] work”.*(Boston store funder)

Chicago, DC, and Minneapolis did not have a single champion but a group of individuals they consulted to inform their business and community engagement decisions. This entity was a non-profit for Chicago and DC, an external CBO and a parent non-profit board, respectively. For Minneapolis, the co-op owners served this role.

The importance of a store champion was particularly showcased in Baltimore when the original store champion left and their replacement did not have the same level of commitment to the store,
*“[The store] was basically left to its own devices and there was no, no general oversight above the manager”.*(Baltimore store leader)

This void ultimately left the store directionless and unable to navigate the many challenges healthy community stores face, lacking a leader to seek out funding opportunities, bring partnerships to fruition, or refine business operations to bring healthy, affordable foods to the community.

#### 3.2.2. Using Nontraditional Business Strategies

Some stores implemented nontraditional business strategies as part of their dual pursuit of financial and social responsibility. For example, Baltimore and Boston adopted a non-profit financial model to allow the stores to offer healthy foods at lower prices. Because of their non-profit status, both stores only needed to cover costs rather than generate profits, and they were ultimately two of the three stores with healthy food prices below the national average. Boston further leveraged its non-profit status and store founders’ connections to obtain philanthropic donations to cover 30% of costs and to procure discounts from vendors aligned with their mission. In the end, Baltimore provided relatively affordable prices but was still unable to reach their breakeven point and closed.

DC considered their store a social enterprise, using a for-profit financial model but incorporating a non-profit parent organization. This financial model facilitated diverse funding streams for the business to remain viable, and some funding was specifically allocated to increase healthy food access. The non-profit director explained the benefit of the DC model to the business and community,
*“…we have some funders who are non-profits who fund directly into the store, we have others who fund into [the parent non-profit] with the provision that those funds get passed directly through to the store, and then we have a few that are funding [the parent non-profit] in order to support [the parent non-profit’s] activities directly”.*(DC store leader)

Minneapolis utilized a co-op model in which community members collectively owned and democratically governed the store. Community members purchased store stock, making them co-owners with influence over store decisions. While originally focused on local, sustainable, and ethically sourced foods, in recent years co-op owners increasingly acknowledged the importance of affordability. Even though store prices remained high at the co-op, Minneapolis store leadership summarized the store’s transition,
*“…[letting] go some of our other [natural food] product standards in order to be more accessible in our community… have a more affordable option”.*(Minneapolis store leader)

Most stores (Baltimore, Boston, Buffalo, DC) took lower than typical profit margins, below 30%, with the goal of offering healthy foods at prices the community could afford. Only Boston seemed to strike the right pricing balance within their business model; although Baltimore prices were also below the national average, pricing barely covered manufacturing costs and delivery costs, creating a very small margin that likely contributed to store closing. However, considering prices in the local context and amongst similar types of retailers was also important. Some stores conducted regular price comparisons with nearby retailers on a core set of food items to ensure their prices were competitive locally. A Boston employee explained the considerations for setting prices,
*“We always want to be lower than those comp prices, but how much lower we are kind of depends on the cost, and you know how essential we think it is…like how much we think people want to see it in the store”.*(Boston store employee)

An alternate model, Minneapolis had overall higher profit margins (40%) and higher price points compared to nearby grocery stores but made pricing more accessible for customers with limited economic resources by pricing staple foods lower (30% margins) and offering a 10% needs-based discount to anyone who participated in any government assistance program or asked to use the program. Ultimately, this needs-based discount made up less than 1% of total sales, reflecting an underutilization of the program by those it is meant to reach.

#### 3.2.3. Obtaining Innovative External Funding

Even with creative business strategies, most stores in this case study relied on innovative external funding to support their business and community, including grants and philanthropic funding. Boston and DC collaborated with CBOs to obtain grants to improve community healthy food access. DC used a grant to subsidize cooking class costs that allowed them to be able to offer free nutrition education to their community, without detracting from their bottom line. Both stores obtained funding to pilot free grocery delivery programs to increase healthy food access to and sales from community members who might otherwise be unable to get to the store, such as people who are elderly or have disabilities.

Additionally, some stores received city and state development funds to improve store infrastructure as well as open additional locations. Buffalo used state funds to expand the size of the store and to prioritize renovations that could increase their healthy food offerings, such as cooling and display cases for fresh produce. DC utilized city funds to expand to an entirely new location and to provide healthy food options to an additional community.

Other stores relied on philanthropic funding to help cover costs directly to the non-profit store (Boston) or through a parent non-profit (DC). This funding was especially critical for Boston in early stages of developing and refining a store model and operations that served the community and remained economically viable. A key funder stated,
*“There were probably at least one, maybe two points, in our funding cycle and [store’s] sort of life cycle, where, you know, I felt, and I think [founder] would agree, you know, but for us [the foundation], [the store] would not still exist”.*(Boston store funder)

Conversely, Baltimore sought to diversify funding beyond their single charitable organization but was ultimately unsuccessful. A Baltimore stakeholder cited the difficulty of raising money for a non-profit grocery store that underscored the importance of leadership that believes in and can generate additional buy-in for a healthy community food store,
*“I’m not asking them to fund a food pantry where we’re giving food away. I’m asking them to fund a business, and so when people put money into a business, they expect a return generally on that investment. There isn’t one”.*(Baltimore store leader)

#### 3.2.4. Using a Dynamic Sourcing Model

Boston, DC, Detroit, and Minneapolis used many vendors and unique sourcing strategies to be able to offer healthy foods at more competitive prices. These stores sourced from a combination of national and local vendors, with DC and Minneapolis putting additional emphasis on local foods. For example, DC built relationships with over 40 vendors to be able to offer a variety of local produce at their store. Additionally, Minneapolis made annual commitments to a dozen local farmers to source upwards of 80% of the co-op’s produce from them in the growing season because, as a store leader explained, they wanted to give,
*“…reassurance to the farmer and with us that we’ll be buying that product, and we can count on them to deliver that product”.*(Minneapolis store leader)

Boston sourced from nearly as many vendors but with the primary goal of lowering prices through nontraditional sourcing, including gleaned produce, donated products, and opportunity buys. Opportunity buys occurred when local and regional vendors needed to move products quickly and included short-coded items, such as a product less than six months before its sell-by date, slow-moving stock, and some rejected items. A Boston employee walked through an example of an opportunity buy that facilitated increased healthy food availability at very low prices,
*“This logistics company called us, and they said they had 300 cases of bags of snap peas. And [a supermarket chain] had rejected them because they didn’t quite like the quality… so I started asking the questions like ‘what’s the code date? what do they look like? what can we retail these for? what do they sell for elsewhere?’ These are like $3.99 you know, they’re 12 oz bag of snap peas, that you would see at [a supermarket chain]. And so we took them; we sent one pallet to each store and we’re selling them [for] 49 cents. And they had like two weeks of code on them so that was part of the determining factor to say ‘okay these are getting donated to us, this is not costing us anything;’ they dropped them off”.*(Boston store employee)

Although stores typically perceived bigger as better when it came to purchasing, Boston leveraged their small size and nimble supply chain to take advantage of healthy food purchasing opportunities in a way larger stores could not, a sourcing model that other community stores could explore. This dynamic sourcing model does have associated costs in the form of time and staff resources, with Boston having three full-time employees dedicated to sourcing and procurement across their three locations.

In contrast, Baltimore used a single wholesaler and thus faced higher costs given limited purchasing power from one vendor. Even with a non-profit model and smaller profit margins, Baltimore offered similar prices to nearby for-profit grocery chains. A store leadership member recounted of their single vendor,
*“…the reason they were the only one is because we did reach out to several, and they were the only one that was willing to work with us... All of other independent grocers have more locations so they scaled their pricing way down... They allowed us to order in smaller quantities at a higher [unit] price”.*(Baltimore store leader)

Alternatively, Detroit and Minneapolis both increased their purchasing power in collaboration with similar retailers. Both stores used collective purchasing, Detroit as a member of a local purchasing group of food retail businesses and Minneapolis as a member of a national purchasing group of co-ops. The stores obtained lower prices from wholesalers through the collective purchasing power of their retailer groups, allowing them to pass on their savings to customers.

#### 3.2.5. Implementing Healthy Food Marketing

All stores used product, placement, pricing, and promotion strategies to increase healthy food sales, aligning with their missions. Notably, Boston only sold healthy products that met nutrition guidelines that were developed in consultation with nutrition experts. Store staff and vendors followed the retail nutrition guidelines for sourcing and stocking foods, and store kitchen staff followed prepared food nutrition guidelines to offer healthy, ready-to-eat meals to compete with nearby fast-food alternatives.

Most stores offered products that widely ranged in their healthfulness but used marketing strategies to promote healthier options. For example, Baltimore had a large produce section near the store entry. DC also placed produce at the store entrance as well as at eye level. In addition to strategic placement, Boston and Minneapolis also promoted healthier options with signage, explaining produce origins and tips for preparation. Buffalo promoted fruits and vegetables by offering convenient, value-added products. As the Buffalo store owner described,
*“You don’t see beer, or you don’t see candy, what we do is put the cooler [of fruit cups and fresh produce] in front of you so the people see it right away”.*(Buffalo store owner)

Most stores offered additional resources beyond that of a traditional grocery store to promote healthier options, including cooking and nutrition education classes. Multiple stores (Boston, Buffalo, Chicago, Detroit) implemented existing programs in partnership with CBOs to implement healthy product, placement, pricing, and promotion strategies. Boston and Detroit participated in Double Up Food Bucks programs that provided an incentive to purchase additional fresh produce when paying with SNAP benefits. Detroit also offered healthy grocery store tours through Share Our Strength’s Cooking Matters program. Both Buffalo and Chicago participated in the Healthy Corner Store Initiative, a national program to increase healthy food availability and promotion at corner stores.

Despite participation in the Healthy Corner Store Initiative, Chicago still used some traditional, unhealthy merchandising strategies, such as placing candy at checkout. The store owner relied on impulse purchases to drive store sales, knowing that,
*“When you walk into a corner store, like mine. You will find all the junk food a front…because those items are our items that are proven that are … well known for being an impulse buy”.*(Chicago store owner)

Baltimore intended to collaborate with CBOs, but ultimately the partnership and programming was not implemented.

#### 3.2.6. Engaging the Local Community

Stores most successful at providing healthy, affordable foods demonstrated continuous engagement with the surrounding community at all levels (e.g., customers, community organizations, employees) and often from the store’s inception forward. Before even opening their doors, successful stores assessed community need and obtained community buy-in on store location. Store owners who were community members themselves, such as Buffalo, had an inherent sense of need and a desire for a store to increase healthy food access. A local Buffalo stakeholder said of the owner,
*“…they feel that he is part of the community. And so, people are very protective, very supportive... And so, he set up shop where others would not, and people appreciate that”.*(Buffalo store stakeholder)

The co-op model at Minneapolis provided a formal process for community members to have a seat at the table to voice effective strategies for improving healthy food accessibility and community reach. A Minneapolis store leader stated,
*“We are a community-owned organization. Our owners are not investors, they are community members”.*(Minneapolis store leader)

Community input on store location was a critical factor for store success. Considering community need, four stores were in USDA-designated food deserts and three stores were in areas considered low-income but not low-access. Stores with leadership from outside the community worked with CBOs and health care systems, tapping into their existing community knowledge and relationships to help identify a location that could increase healthy food access and reach community members with limited economic resources. In contrast, Baltimore chose a store location based on operational convenience for their parent organization rather than community need or input. Baltimore store leadership explained,
*“…the store was never placed in a food desert, so there was no overriding reason to drive people to the store because they could go to a [grocery store chain], the [grocery store chain] was half a mile from our store”.*(Baltimore store leader)

Additional detail on the depth of community engagement and how it varied across stores is further described in Kaur et al., 2022 also in this special issue [46].

### 3.3. How Did Stores Balance Missions and Margins to Achieve Success?

At the most basic level, six of the seven stores featured in this case study were successful in remaining operational businesses. More specific to healthy community food stores, we found three main criteria for success: (1) making a variety of healthy foods available, (2) offering these foods at affordable prices, and (3) reaching community members with limited economic resources. Boston was the only store to clearly achieve all components of success. DC, Detroit, and Minneapolis were close to meeting the three-pronged definition of success, but did not excel in one of the key areas. Baltimore met only one criterion, affordability, and ultimately closed.

Tensions to balance mission and margins existed across all three facets of healthy community store success. To make healthy foods available, stores navigated their role to meet community demands versus change them. To make healthy food affordable, stores balanced offering prices that were accessible to their community but still generated adequate revenue for their business to remain operational and, in some cases, supported a fair and sustainable food system. To reach community members, stores explored if origins within the community were necessary or if external leadership paired with authentic community engagement could prove effective.

We also identified six strategies for success: (1) having a store champion, (2) using nontraditional business strategies, (3) obtaining innovative external funding, (4) using a dynamic sourcing model, (5) implementing healthy food marketing, and 6) engaging the local community. Three stores, Boston, DC, and Minneapolis, implemented all six strategies for success. Baltimore implemented just two strategies and ultimately closed.

Detroit was relatively successful yet only implemented four of the six strategies. Importantly, Detroit’s community context enabled success as it had been the only grocery store in the community for decades. Additionally, as a supermarket, Detroit did not need to rely on a store champion or novel business strategies because its store type and size more easily facilitated the stocking and the pricing of an array of healthy foods at affordable prices relative to smaller retailers. Smaller stores needed to be more intentional and creative with healthy food sourcing, considering their limited shelf space and purchasing power. In addition to store type, store strategies varied by store financial model. Non-profit stores could more easily align business strategies with their mission given their goal of reaching breakeven, not generating revenue. Boston’s non-profit model with philanthropic funding enabled them to pay staff dedicated to establishing and operating their complex sourcing model, absorb potential loss from not selling unhealthy foods, and still be able to offer healthy foods at low prices.

On the other hand, two of the stores that implemented all six strategies did not achieve all components of healthy community food store success. DC and Minneapolis adopted all strategies identified in the paper, but their more expansive missions led to business decisions that led to higher healthy food prices. For example, both stores prioritized sourcing local food to support their local economies and had to pass on the associated additional costs to customers. Minneapolis had the highest healthy food prices of all stores, necessary given their vast mission that spanned economic, social, and environmental responsibility; their mission mandated prices that reflected the true cost of food.

## 4. Discussion

This is the first study to use a case study approach to examine strategies for success in a diverse sample of community grocery stores. We used cross-case analysis to identify a multi-faceted definition of healthy community store success encapsulating healthy food availability and affordability as well as community reach. Subsequently, we uncovered key strategies for success that help stores balance missions and margins, spanning store leadership, business strategies, external funding, sourcing models, marketing, and community engagement. By gaining detailed insights into these stores, we may identify how other food retailers could increasingly move toward offering affordable, high-quality, healthy foods in communities that disproportionately lack access.

This study adds a comprehensive definition of healthy community store success to the literature that builds upon existing evidence. Previous research has used the Nutrition Environment Measures Survey in Stores (NEMS-S) to evaluate the availability of healthy foods and beverages in areas with economic disadvantage in a single community, city, or state [35,36,37,38,51,52,53,54,55,56]. Some of these studies considered both healthy food availability and affordability using NEMS data, however the store samples were not limited to community stores [53,54,55,56]. Previous research of healthy retail interventions has also assessed both healthy food availability and community engagement in specific localities, including in the evaluations of the Wapuca Eating Smart, Healthy2Go, Baltimore Healthy Eating Zones, and Healthy Eating, Active Communities programs [28,57,58,59]. This is the first paper to consider a multi-faceted definition of community store success that is national in scope and spans healthy food availability, affordability, and community reach.

Previous research has individually recognized some of the six strategies we identified for success [16,17,18,19,20,21,22,23,24,25,26,27,28,60], but our comprehensive set of strategies associated with successful healthy community stores is novel. The focus of the literature on community store strategies to improve healthy food access has been on healthy marketing, including an array of successful healthy product, placement, price, and promotion interventions in community stores [19,20,21,22,23,24,25,26,27,28,60]. Additionally, research on community store sourcing has established that retailer–supplier agreements influence the store food environment [16,17,18], however specific sourcing strategies to improve the healthfulness of the store food environment, such as number of vendors, local versus national distributors, and leveraging opportunity buys, had not previously been identified. Our study also brings novel attention to the community store leadership, business, and funding strategies associated with increasing healthy food access.

Stores did not need to implement all strategies to be successful but appeared to need to reach a certain threshold, as adopting only two strategies was associated with store closure. Additionally, it may be simplistic to solely consider if a store implemented a strategy and the depth of strategy implementation is also important. Another key differentiator was which strategies were implemented. The leadership of a consistent, long-term store champion and community engagement emerged as critical strategies across stores. Dombrowski et al. also identified community store owners as key health promotion agents and found dual prioritization of people and profits to be a critical factor for success [61]. Most stores in the case study had a store champion that advocated for both the store’s business and community, and they could navigate the unique challenges of a community store. The departure of the Baltimore store champion and subsequent store closure emphasized the importance of a champion, as well as the importance of analyzing cases that ranged in degrees of success. While an essential strategy, a charismatic and a dedicated store leader is not an easily sharable or scalable strategy, and it may not be available in all communities. Of note, Detroit did not have a store champion, emphasizing that which strategies were necessary varied by store context and type. Future case studies and highly contextualized research could more deeply explore store decisions that optimize success for balancing mission and margins.

This paper finds community food stores can increase healthy food access while sustaining a viable business, with research, private sector, philanthropy, and policy implications. Small retailers play an important role in communities with limited economic resources [4] yet are understudied due to data limitations [15]. This paper demonstrates the case study approach can be appropriate for this setting, especially considering the unique contexts of independent stores. Additionally, the study provides novel definitions (e.g., healthy food store success) and metrics (e.g., Healthy Food Availability Healthy Eating Index) that can be utilized in future research to generate more consistent outcomes in this field with easier comparisons across studies. Additionally, this study identifies strategies community food stores can adopt and broader food retailers can adapt for their store environment. The paper also identified the importance of external funding and community partnerships. Foundations and non-profits can support retailer strategies identified in the study through co-implementing and funding healthy retail interventions. The study results can also inform state and local policies that facilitate retailer prioritization of healthy food access through grants, loans, tax credits, equipment, and technical assistance, such as Illinois’s Healthy Food Program Development Act [62].

This study had strengths and limitations worth noting. The study used a maximum variation sampling approach to select seven stores with considerable variation across dimensions of interest, including store type and business model. Additionally, the mixed-methods data collection and the highly contextualized analytical approach were well suited to the research question at hand and the store environment of interest. Data collection from multiple data sources and multiple perspectives provided within-case triangulation and research team discussions across the multiple sites and research teams provided cross-case triangulation; although, these were limited to those within the larger study team. Other limitations included the store concentration in the Northeast and the Midwest regions of the United States. Future research could benefit from exploring stores from a wider geographic range, including sites at different levels of urbanicity (e.g., suburban areas, rural areas). Additionally, the study did not include the full range of store success with only one store failing. Purposively sampling more deviant cases could have made clear additional strategies and contexts that enabled success. Future studies could also incorporate detailed sales outcomes where technology, such as point-of-sale systems, are available or outcomes of ultimate interest, such as food consumption. Future research could build upon this case study, including customer surveys with rigorous dietary intake assessment measures to strengthen understanding of healthy community store success and achievement of a store mission to improve dietary behaviors in the communities they serve.

## 5. Conclusions

This paper identified what healthy community store success looks like and strategies for how stores achieved it. Healthy community stores did not need to implement all strategies for success, however certain strategies, such as a store champion, emerged as especially critical. Additionally, some stores adopted all strategies but did not achieve all components of success due to expansive missions that necessitated higher food prices. To be successful, stores had to balance their mission and margins, carefully considering how to make healthy food available at prices people could afford to reach members of their community while sustaining a viable business. Retailers can commit to healthy food access and adopt and adapt these strategies for success, especially if policymakers and foundations can incentivize healthy retail adaptations.

## Figures and Tables

**Figure 1 ijerph-19-08470-f001:**
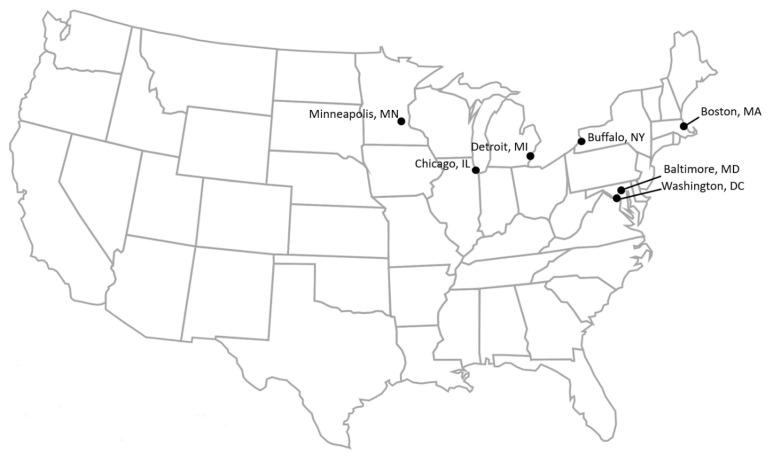
Healthy community store locations.

**Table 1 ijerph-19-08470-t001:** Healthy community store characteristics ^1^.

Store Location	Operating Dates	Financial Model	Store Type	Store Size
Baltimore, MD, USA	2018–2021	Non-profit	Grocery store	7000 sq ft
Boston, MA, USA	2015–present	Non-profit	Grocery store	3850 sq ft
Buffalo, NY, USA	2007–present	For-profit	Corner store	--
Chicago, IL, USA	2003–present	For-profit	Corner store	3500 sq ft
Detroit, MI, USA	1984–present	For-profit	Supermarket	65,000 sq ft
Minneapolis, MN, USA	1970s–present	Co-op	Grocery store	20,000 sq ft
Washington, DC, USA	2014–present	For-profit	Market	900 sq ft

^1^ Adapted from Gittelsohn et al., 2022 [31].

**Table 2 ijerph-19-08470-t002:** Healthy community store success: intention, action, and achievement by store.

	Stores						
Components of Success	BAL	BOS	BUFF	CHI	DC	DET	MINN
*Store commitment to healthy food*							
Store acknowledged importance of healthy food (Intention)	√	√	√	√	√	√	√
Store defined “healthy” (Action)		√			√		√
Store defined “healthy” food (Action)		√					
Store operationalized “healthy” (Action)		√	√		√		√
Store provided a wide variety of healthy foods ^1^ (Achievement)		√			√	√	√
*Store commitment to affordable prices*							
Store acknowledged importance of affordable prices (Intention)	√	√	√	√	√	√	√
Store altered business model to offer affordable prices (Action)	√	√	√		√	√	√
Store provided affordable prices ^2^ (Achievement)	√	√				√	
*Store commitment to reaching community with limited economic resources*							
Store acknowledged importance of reaching community (Intention)	√	√	√	√	√	√	√
Store engaged with community (Action)		√	√	√	√	√	√
Store customer base reflected community ^3^ (Achievement)		√	√	√	√	√	√

^1^ Store had a Healthy Food Availability Healthy Eating Index (HFAHEI) score above 20, within 5% margin of error. ^2^ Store sold healthy basket of goods (low-fat gallon of milk, pound of apples, pound of tomatoes, pound of chicken breast, and loaf of whole wheat bread) for less than the Consumer Price Index (CPI), within 5% margin of error. Cost was adjusted if item(s) were not sold (Buffalo, DC, and Chicago did not sell low-fat milk; Buffalo did not sell chicken breast). ^3^ Store customers’ economic resources reflected community economic resources, including nutrition assistance program participation, Section 330 residential status, and/or key informant perceptions.

**Table 3 ijerph-19-08470-t003:** Healthy community store success: strategies implemented by store.

		Stores						
Strategies	Examples	BAL	BOS	BUFF	CHI	DC	DET	MINN
Had a store champion	Store owner(s)	x	√	√	√	√		√
	Store founder							
	Non-profit board							
Used nontraditional business strategies	Non-profit, co-op model Lower profit margins Needs-based discounts	√	√	√		√		√
Obtained innovative external funding	Grants City development funds Philanthropic funds		√	√		√	√	√
Used a dynamic sourcing model	Sourced from a variety of vendors (wholesaler, farmer, gleaner) Opportunity buys Collective purchasing		√			√	√	√
Implemented healthy food marketing	Stocked healthy *products* Prominent healthy item *placement* *Price* discounts on healthy items *Promoted* healthy products	√	√	√	√	√	√	√
Engaged local community	Input on store location Collaborated with CBOs Offered engagement activities (cooking classes, yoga)		√	√	√	√	√	√

## Data Availability

The data presented in this study are available on request from the corresponding author. The data are not publicly available due to confidentiality considerations.

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
