# Peer review of "Balancing Mission and Margins: What Makes Healthy Community Food Stores Successful"

_ijerph, 2022, doi:10.3390/ijerph19148470_

Round 1
Reviewer 1 Report
Dear authors,
with great interest I read your manuscript and I appreciate the content and methodological approach. However, I few comments that may help you to improve the manuscript.
P2, L 59-66: It would be good if the reader gets a short overview on the landscape of community stores in the US--> Different foci, for instance local, organic, natural food sourcing, justice focus--> affordabilitity --> healthy food for everyone, particular the food poor. Also elaborate on the criteria that these store commonly considered for their missions and targets. That is important since in your problem you bring up strategy. Here I will expect that you comment on residents, community, assets, incentives and how the surrounding community is reflected in the store.
P2, L. 74-80: Problem statement: You indicate you would like to explore what makes these community health stores succesful or not? Succesful in which sense. Please elaborate. Please also add what the reader gains from knowing about the unsuccesful store. To strenghen the value of your research outline where within the existing body of literature your work is embedded.
P. 2 L. 81: A literature review is completely missing. This needs to be added.
P2. L 90: A multiple case study approach is more than appropiate, however the current justification is weak. Add explanation why your chosen approach to follow Gittelsohn et al. (2022) is most appropiate. Maybe you want argue and consider Eisenhardt or Yin as points of comparison when it comes to multiple case study.
P2. L.95: Can you please name and justify your chosen sampling approach/ approaches. The practical details seem fine. But the reader should be able to put your choice in perspective.
P3. L 106: Multiple data sources is fine and good practice when it comes to case studies. Did the authors use other forms of triangulation other than data triangulation. Please elaborate. Can the authors please elaborate what has been done to assure rigor within the casy study. Which criteria where considered and why?
Results and discussion chapter: I suggest that will be merged. So findings are directly put in perspective with the recent body of literarture. In the current stage it tedious to read. After on text excerpt it switches to the next aspects and I am asking myself so what. The text excerpts should also put into perspective- certainly anonoumsly, but the reader should have a rough idea- where excerpt comes from. Without information related to the sources where they stem form this does support authenticity and does give your research participants a voice.
P. 15. L 659: Good acknowledge to limitation but no need to trash your own work. I find the comments on validity and generlizability specifically bad. Maybe reconsider and soften a bit.
I hope this helps. I am looking forward to see the revised manuscript
Reviewer 2 Report
The authors have explored an important issue in this paper and have provided insightful results; however, the paper needs some work before it can be published.
To start, this paper is not situated in existing literature. What academic literature has been written on this topic, and what is the existing theoretical, methodological, or empirical research gap that needs to be filled? This remains unclear. The start of the paper is clear but underdeveloped. Additionally, concepts such as food access, healthy eating, and food system need to be further examined as there are many different understandings of what these terms mean in the field at large.
In the methods section, more could be added to provide further detail on the rationale, process, and strengths/weaknesses for choosing the various stores/locations.
In the results section, more could be added to distinguish the various data by location/case study in order to highlight what was found (describe for the reader) and why it is important for the study and existing literature (analyze the significance for the reader)
In the end of the paper, the authors could explain the significance of the findings for the reader at a broader level. In particular, why are the findings significant for existing research, future research, and policy? I think it is important for this paper to more directly speak to academic literature, as it reads too much as a report right now.
Other Items to Consider
-Add more graphics, including possibly a conceptual road map, methodological flow diagram, and location map
-Read through the paper for clarity and writing quality
Round 2
Reviewer 1 Report
I enjoyed reviewing this paper and was excited to see the improvements made. I think you are good to go, I have no more reservations towards the manuscript. Well done!
Reviewer 2 Report
The authors have done a good job responding to the reviewers' comments. For this reason, this paper is ready for publication.